# Managing Building Water Disruptions in a Post-COVID World: Water Quality and Safety Risk Assessment Tool for Academic Institutions and School Settings

Stephanie C. Griffin [1,†], Molly M. Scanlon [2,*,†] and Kelly A. Reynolds [2]

1    College of Health Sciences, Stockton University, Galloway, NJ 08205, USA
2    Department of Community, Environment and Policy, Mel and Enid Zuckerman College of Public Health, University of Arizona, Tucson, AZ 85724, USA
*    Correspondence: mscanlon@arizona.edu
†    These authors contributed equally to this work.

**Abstract:** Fluctuating building occupancy during the COVID-19 pandemic contributed to poor water quality and safety conditions in building water distribution systems (BWDSs). Natural disasters, man-made events, or academic institutional calendars (i.e., semesters or holiday breaks) can disrupt building occupant water usage, which typically increases water age within a BWDS. High water age, in turn, is known to propagate poor water quality and safety conditions, which potentially exposes building occupants to waterborne pathogens (e.g., *Legionella)* associated with respiratory disease or hazardous chemicals (e.g., lead). Other influencing factors are green building design and municipal water supply changes. Regardless of the cause, an increasing number of water management policies require building owners to improve building water management practices. The present study developed a Water Quality and Safety Risk Assessment (WQSRA) tool to address gaps in building water management for academic institutions and school settings. The tool is intended to assist with future implementation of water management programs as the result of pending policies for the built environment. The WQSRA was modeled after water management practices created for controlling water contaminants in healthcare facilities. Yet, a novel WQSRA tool was adapted specifically for educational settings to allow building owners to evaluate risk from water hazards to determine an appropriate level of risk mitigation measures for implementation. An exemplar WQSRA tool is presented for safety, facility, industrial hygiene, and allied professionals to address current gaps in building water management programs. Academic institutions and school settings should examine the WQSRA tool and formulate an organization-specific policy to determine implementation before, during, and after building water-disruptive events associated with natural or man-made disasters.

**Keywords:** commissioning; disasters; green buildings; *Legionella*; risk assessment; schools; sustainability; water disruption; water management; water quality

## 1. Introduction

Universities and school systems commonly experience fluctuating building occupancy due to the academic calendar year (e.g., summer break, winter holiday, or spring break) [1–3]. The academic calendar typically creates periodic weeks or months with low or no flow water levels in a building water distribution system (BWDS). However, the COVID-19 pandemic stay-at-home orders exacerbated issues related to water disruption and poor water quality conditions at university and school campus facilities [1–5]. During the pandemic, students were sent home, while some institutions maintained limited working hours for faculty and staff [4]. Drinking fountains and bottle fillers in many buildings were labeled as no-use to reduce the likelihood of surface transmission of the SARS-CoV-2 virus [2]. Initially, there was minimal awareness of how low- or non-use of the BWDS would reduce water quality and safety. As the pandemic progressed, the World Health Organization (WHO) [6], the United

States (US) Centers for Disease Control and Prevention (CDC) [7], and the US Environmental Protection Agency (EPA) [8] issued warnings and guidance documents for all building owners about the need for managing water quality due to the high number of unoccupied buildings. These public health organizations recommended implementation of a water management program (WMP) for controlling microbial risk (e.g., *Legionella*, *nontuberculous mycobacteria* (NTM), or *Pseudomonas*) as well as leaching of metals (e.g., lead and copper) in the BWDS [6–8]. *Legionella* and other waterborne pathogens are likely to grow and spread in the BWDS with increasing water age (e.g., water dormancy, low flow or no flow conditions), low residual disinfectant, and poor temperature control (i.e., bacterial growth within permissive ranges) for both hot- and cold-water systems [9]. *Legionella* is a waterborne pathogen associated with respiratory illness in humans known as legionellosis or Legionnaires' disease (LD) from the 1976 outbreak near a convention center in Philadelphia, Pennsylvania [9,10]. Physical and chemical changes in the BWDS (e.g., soil and sediment invasion, flow velocity) can cause pipe corrosion and contribute to poor water quality, leading to high levels of lead or copper impacting safe drinking water [2,9].

Similar to the impacts observed during the COVID-19 pandemic, academic institutions and schools are located in community settings that are increasingly vulnerable to surrounding environmental conditions and disaster events. Water quality and disaster events are often interchangeably linked [11]. As we have seen with the Flint, Michigan [12] and Jackson, Mississippi [13,14] water crises, as well as the East Palestine, Ohio, train derailment [15,16], any community and its local educational system can be critically impacted based on compromised water resources or infrastructure. As background analysis for the present study to create a tool for water quality risk assessment, we reviewed five key areas related to educational settings, including (1) water-disruptive events; (2) green building design and water system challenges; (3) municipal water supply changes; (4) water management policies for BWDS; and (5) risk decision matrix concepts.

### 1.1. Disruptive Water Events

Water is often referred to as the invisible infrastructure; it operates safely and efficiently until there is a system failure [17]. Disruptive water events occur during natural disasters and man-made events [18,19]. These events can include but are not limited to reduced occupancy or stay-at-home orders during a communicable disease outbreak [1,4,20], excessive rainfall and flooding [21,22], deteriorating infrastructure , and construction activities [19]. The disruptive event impacts BWDS performance, which, in turn, impacts water quality and safety and results in an increase in the likelihood of water-related disease cases, injury, or death [10,20]. The CDC identified these events as "unmanaged external changes," which contributed to 35% of the LD cases during their North American field investigations from 2000 to 2014 [18]. In August 2018, Bresso, Italy, experienced heavy rainfall and increased relative humidity [22]. The subsequent epidemiological investigation reported a high concentration of disease cases near the public water fountain in a community-acquired legionellosis outbreak of 52 disease cases, including 5 deaths. Similarly, construction activities have been associated with disease cases and deaths from *Legionella*, NTM, *Sphingomonas*, and *Fusarium* in both healthcare and community settings [19].

### 1.2. Green Building Design and Water System Challenges

Green building design (GBD), an increasingly popular element of educational facility design and operations [23,24], has been associated with water efficiency initiatives that are seldom checked to verify safe operations [9,12,19,25,26]. GBD has developed a difficult reputation in the water management industry. The challenges created by GBD features are unintended consequences of design professionals and building owners implementing energy conservation building rating and certification systems, such as the US Green Building Council's (GBC) Leadership in Energy and Environmental Design (LEED), without consideration of the potential impact on water quality and safety parameters (e.g., disinfectant residual, water age, water temperature, or other). Green potable water systems are awarded

certification points when water demand is reduced by 20–50% [12,26]; however, the BWDSs are often designed for maximum building occupancy usage. Waterborne pathogens can flourish under BWDS conditions with high plumbing surface area-to-volume ratios, especially when variable water flow conditions occur. LEED building water conservation initiatives have developed a negative connotation within the water management industry. The LEED acronym has been unfairly targeted and renamed as *Legionella* Enabled Engineering Design [9]. Rather than stigmatize the building rating system, the National Academies of Science, Engineering, and Medicine (NASEM) [12] encouraged a review of such criteria in the context of implementation of a WMP to encourage a balance of water efficiency with water quality and safety to achieve a sustainable water supply system (see Figure 1).

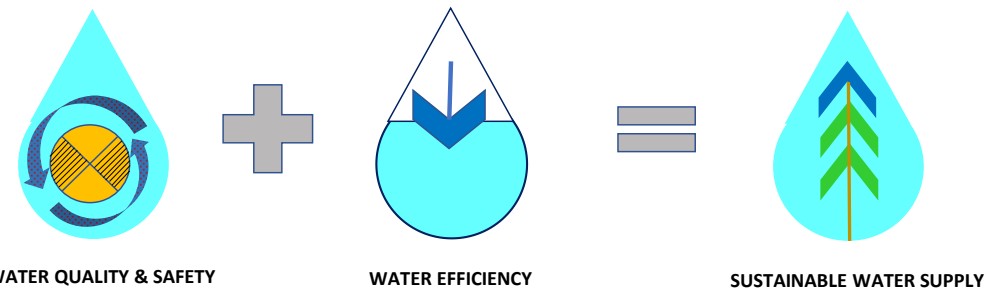

**WATER QUALITY & SAFETY**          **WATER EFFICIENCY**          **SUSTAINABLE WATER SUPPLY**

**Figure 1.** Conceptual framework of achieving a sustainable water system.

### 1.3. Municipal Water Supply Changes

Changes in municipal water systems and their relationship to building owner management of water systems in both community and institutional settings must be carefully considered. In the renowned case study of Flint, Michigan, the Michigan Department of Environmental Quality approved a water source switch from Lake Huron water provided by the Detroit Water and Sewerage Department (DWSD) to the Flint River water and plant in April 2014 as a cost-saving measure [12]. Within 30 days, residents complained of poor water quality. Within 60 days, outbreaks of LD cases emerged in elevated numbers within the water service area. Over the next two years, 86 disease cases emerged in the region (2014, $n$ = 42 cases; 2015, $n$ = 44 cases) [12]. In the summer of 2016, after Flint switched back to DWSD-treated water from Lake Huron, LD cases returned to pre-2014 levels. These events also led to a US Federal Emergency declaration by the Obama administration and investigations into elevated lead exposure among Flint residents, especially young children. Similarly, in 2015, 58 residents of the Illinois Veterans Home Quincy (IVHQ) developed LD and 12 died from exposures that were initially assumed to be isolated to an IVHQ central water heater storage tank taken offline for maintenance and returned to service [21]. However, those maintenance activities do not explain the coincident five additional community cases occurring among occupants of single-family residences with no overlapping exposure with the IVHQ campus or buildings. Rhoads et al. [21] expanded the research investigation and stated that many changes to municipal water treatment and distribution of water are under-regulated and not effectively communicated to the public and customers, including BWDS managers. The authors hypothesized that high levels of rainfall and flooding, alterations to water treatment, and interruptions to corrosion control likely caused a continuous decrease in disinfectant residuals throughout the underground water distribution system, contributing to increased lead levels and *Legionella* growth and spread.

### 1.4. Water Management Policies for BWDS

To control for water quality and safety within a wide variety of disruptive water events, NASEM [12] recommended that in addition to the emphasis on healthcare facilities, all public buildings (e.g., schools, apartments, hotels, commercial businesses, and government buildings) implement a WMP to raise the standard of care for waterborne pathogen hazard analysis and control in BWDSs. The current water management industry guidance, standards, research, practice, and tools tend to focus on (1) healthcare building

types, since patients with underlying disease or immunocompromised health status are at higher risk for respiratory diseases associated with waterborne pathogens (e.g., *Legionella*); and (2) ongoing operations after the building is functioning. A portion of the healthcare emphasis stems from a previous 2017 US federal agency mandate: the US Centers for Medicare and Medicaid Services (CMS) [27] required the implementation of a WMP for healthcare facilities to reduce the likelihood of waterborne pathogen healthcare-associated infections. US agencies referred building owners to develop WMPs in alignment with guidance documents from the American National Standards Institute (ANSI) and the American Society of Heating, Refrigerating, and Air-Conditioning Engineers (ASHRAE) Standard 188: Legionellosis Risk Management for Building Water Systems [28], as well as the US CDC Toolkit: Developing a Water Management Program to Reduce *Legionella* Growth and Spread in Buildings [29]. For other building types (e.g., schools, hotels, or commercial structures), WMP compliance is mostly voluntary and not yet required without additional state or federal legislation [12]. Although these same standards include requirements for managing water disruption and performing a risk assessment, there is minimal guidance and tools on how a water management team should anticipate, recognize, evaluate, and control such disruptive water events.

As the burden of disease from water contaminants increases [30], water management policies have advanced and will impact academic institutions and schools. The CDC estimates the US population is impacted by specific waterborne pathogens each year (i.e., 6630 deaths, 118,000 hospitalizations, and 7.15 million illnesses) contributing to USD 3.33 billion in annual healthcare costs [30,31]. Prior studies of a broader range of waterborne pathogens and adverse health outcomes estimated up to 19.5 million cases of illness each year [32]. Approximately 94% of waterborne pathogen-associated deaths were from biofilm in water distribution piping (e.g., municipal water infrastructure or BWDS) [30,33]. *Legionella* is but one waterborne pathogen of interest, and yet its impact is significant and increasing; since 2000 the CDC has seen a nine times increase of LD cases [34]. During the COVID-19 pandemic, schools and universities have experienced poor water quality resulting in both microbial [1,3,5] and chemical exposures [2,4].

Since the COVID-19 pandemic, several states have moved legislation forward requiring building owners to develop WMPs. California [35], Illinois [36], and Pennsylvania [37] have proposed or passed legislation to require various building types and building owners to comply with WMP standards. Additionally, New York has had extensive legislation on WMP implementation including cooling towers since 2017 [38,39]. Furthermore, some anticipate more water legislation is forthcoming in order to access the USD 111 billion federal water funding following passage of the 2021 US Bipartisan Infrastructure Bill [40,41]. The infrastructure bill is intended to improve access to clean drinking water service lines to millions of American households, businesses, and schools. Water infrastructure legislation of this magnitude will likely have tremendous impact on many measures of water quality and safety to rebuild these systems. Yet, there are minimal tools available to assist the over 5900 US colleges and universities and the 128,000 US elementary and secondary schools [42] with any sort of water quality and safety risk assessment for water-disruptive events, including infrastructure and construction modifications to the BWDS.

### 1.5. Risk Decision Matrix and Water Management

A risk decision matrix (RDM) is a basic tool used to evaluate environmental hazards in a qualitative or semiquantitative decision-making process to (1) transparently arrive at a common understanding of stakeholders of an adverse event, and (2) categorize and prioritize risk and determine an appropriate course of action [43,44]. A RDM is typically depicted using a graphic chart in which consequence (x-axis) is evaluated against likelihood (y-axis) to determine an outcome (z) that aligns with a level of risk or risk mitigation for the hazardous condition under evaluation [43–45]. The RDM's graphic chart can use the simple format (3 × 3 cells) or increase in range to the more complex (5 × 5 cells) format and variations in between (see Figure 2) [17,43–45]. The RDM chart's cell structure is

often organization- or industry-specific. Industries known to frequently use RDM methods include aerospace, construction, medicine, transportation, mining [45], and industrial hygiene [46,47]. By establishing risk mitigation levels (RMLs), users can implement a proposed set of risk mitigation (e.g., hazard controls) for an identifiable hazard or hazardous condition. Over time, RMLs can be re-evaluated to demonstrate the effectiveness of a control [45].

**Figure 2.** Risk decision matrix conceptual framework illustrating a basic 3 × 3 cell format (white cells) with options for expanding to a 4 × 4 cell format (using added yellow cells) or further expansion to a 5 × 5 cell format (using added red cells).

The same RDM method had been extrapolated for healthcare construction activities [48]. Infection preventionists were faced with evaluating and controlling airborne pathogens (e.g., *Aspergillus*) during construction projects in hospitals [49]. Routine construction projects (e.g., demolition, maintenance and repair, renovations, additions, and new construction) require evaluation and control for both air- and waterborne pathogens that have contributed to healthcare-associated infections (HAIs). The specific healthcare RDM method is commonly referred to as an infection control risk assessment (ICRA) for healthcare construction activities using a 4 × 4 cell graphic matrix [48] (Chapter 2, p. 14). ICRA defined the x-axis of consequence in terms of the size and scope of the construction event (i.e., the construction scope of work) from minimally invasive to full-scale renovation or new construction, using project type letters A, B, C, and D. The y-axis was defined in terms of patient risk groups by location of the building (e.g., offices, lobby, radiology, emergency, intensive care unit, hemodialysis, and other major departmental areas). Although patients could be anywhere in the building, the risk group definitions examined the exposure to airborne pathogens within various departmental building areas, according to the potential susceptibility of patient population harm from airborne hazards. The z outcome was defined as the risk mitigation level (RML) using numeric values articulating a set of hazard controls to be implemented by the maintenance personnel or contractor. Compliance is typically verified by the infection preventionist assigned to the hospital's construction project. Scanlon et al. [50] recently published a novel ICRA for waterborne pathogens for healthcare construction activities pulling together a similar 4 × 4 cell RDM by citing the latest evidence, guidance, or standards used in prevention.

To summarize, although there has been significant worldwide water quality research for utility infrastructure [51], environmental water pollution [52], water hydraulic behavior [53], and municipal water delivery systems for drinking water [54], there are far fewer examples of water research, tools, and standards focused on the building owner and their respective water management team members, who are often the sole decision makers responsible for managing water quality for building occupant usage [19,20,55]. In recommending WMPs for all public buildings, the NASEM report [12] suggested creating derivative tools or products focusing on non-healthcare buildings to further advance water management best practices to a wider building owner audience. The novelty of

this study is represented by the building type, academic institution and school settings, and the simplicity of the tool for implementation during a disastrous event impacting water quality and safety. First, educational settings are a different building occupancy type (i.e., non-healthcare) and need risk assessment tools addressing environments used by children, minors, and young adults, as well as faculty, staff, visitors, and other community members. Second, the building owner's facility management personnel often become involved in a water-disruptive event, given their knowledge about the BWDS physical components (i.e., fixtures, piping, and water heaters), and yet they may not fully understand or be trained in building water management or basic water science. The building owner's representatives need tools to rapidly evaluate the water-disruptive event, establish team consensus, and subsequently communicate instructions to other stakeholders about necessary modifications to the BWDS to re-establish water quality for ongoing operations. The purpose of the current study is to fill this gap in current water management practices by developing a water quality and safety risk assessment (WQSRA) tool that will: (1) assist higher education and school-based water management teams with evaluating water quality challenges at academic campuses and school environments during water-disruptive events, and (2) increase awareness and reduce potential risk to all building occupants (e.g., students, faculty, staff, and visitors).

## 2. Materials and Methods

The proposed WQSRA tool will leverage core principles of water management practice and establish WMP risk mitigation methods for water quality and safety [28,29,56,57] using an RDM. The intent is to create alignment with existing water quality and safety standards and practice to further the development of a comprehensive WMP for building occupancy types other than healthcare to implement before, during, and after water-disruptive events. The methods stated herein outline the development of a WQSRA decision matrix for academic campus buildings and school facilities.

### 2.1. Developing WQSRA Project Types

The water management framework from healthcare construction will be modified for water-disruptive events by focusing on disruption types and reducing the number of categories from four to three (i.e., A, B, and C) similar to other RDMs for water and public health impact [17]. The key parameters involve principles of water age [56] and defining the scope of the water disruption event (e.g., duration of disruption, number of building shutdowns, low occupancy, repair, maintenance, renovation, or newly constructed BWDSs) [50].

#### 2.1.1. Adjusting for Water Age/Dormancy

Water age is a significant risk factor contributing to reducing water quality and safety [9,29,56]. During water-disruptive events, high water age occurs when low- or no-use building occupancy occurs or because of water service disruption. Additionally, during construction activities, high dormancy typically occurs after the BWDS is filled with water and is left stagnant until the building owner has established ongoing day-to-day operating conditions [56]. Within the WQSRA framework, water age (time of dormancy) would need to align with categories A, B, and C from minimally disruptive activities to major disruption.

#### 2.1.2. Defining Disruptive Event and Scope of Work for Water System Components

The categories (A, B, and C) define plumbing and BWDS components based on the water-disruptive event's complexity and invasiveness into the BWDS [50]. For instance, modifications to a shower room are likely less complicated than replacement of a central water heating system (e.g., a boiler in a central utility plant) that impacts the total building area and potentially all building occupants.

## 2.2. Defining Building Area Risk Groups

The WMP team (e.g., facility manager, safety officer, industrial hygienist, among others) will need to categorize risk to building occupants by reviewing a building list or campus facility map including the building types (e.g., athletic facility, classroom, or dormitory) and room functions (e.g., training center, restroom, or food service). Subsequently, the WMP team will identify water fixture types for potential risk of exposure to aerosolized water (e.g., sinks, showers, whirlpools, ice machines, misters, or other devices with a water reservoir) [29]. For example, dormitories or athletic venues with whirlpools or shower facilities are considered higher risk than classrooms with no sinks or running water. Special consideration should be given to any building occupants with known or suspected high-risk exposures (e.g., children's exposure to lead in water) or health status (medical disability, immunocompromised status, or underlying disease status) [28]. A certified or licensed professional (i.e., clinician, infectious disease specialist, or industrial hygienist) should be part of this evaluation process. Final designations for building occupant risk groups are the responsibility of the organization's WMP team.

## 2.3. Developing WQSRA Risk Mitigation Strategies

Risk mitigation strategies are the equivalent of determining appropriate control measures within a WMP. The WMC framework [50] categorized and prescribed a list of control measures that defined a systematic course of action appropriate for a construction project scope and the patient population at risk. Mitigation strategy implementation methods (i.e., hazard control options) are suggested based on water management industry guidance [28,29,56–58], standards [28], and evidence-based practices [12,20] that are subsequently grouped into WQSRA risk mitigation strategies; the technical aspects of these mitigation strategies are described in detail in Scanlon et al. [50] (pp. 345–347) and are summarized in Table 1 below.

**Table 1.** WQRSA Mitigation Strategies.

| WQRSA Mitigation Strategy | Description | Notes |
|---|---|---|
| **Monitoring Residual Disinfectant (Free or Total)** | Chlorine (measured as free residual oxidant, FRO) or monochloramine (measured as total residual oxidant, TRO) are the most common disinfectants [13,19]. | Reduced risk of growth and spread of pathogens in BWDSs with FRO between 0.20 and 4.0 ppm or TRO between 0.50 and 4.0 ppm [12]. |
| **Monitoring and Maintaining Temperature** | Maintain hot water ranges and cold water thresholds to discourage growth of *Legionella* [59]. | Cold water maintained at $\leq$77 °F (25 °C) and hot $\geq$113 °F while also avoiding scald risks [29]. |
| **Flushing Protocols** | Flushing BWDSs helps maintain temperature ranges, reduce water age, and introduce "newer" water with adequate residual disinfectant into the system. Protocols are highly dependent on the volume of water within the BWDS; calculations may be needed to move and replace 100% of the total water volume in response to a disruption event. | Protocols should specify the minutes of flushing, number of days of the week for flushing, and the number of fixtures to be flushed. |
| **Utilizing Filtration** | Used to remove suspended particles from the potable water system before dispensing water at the terminal fixture [56]. Filters, screens, and other devices are commonly applied at the point-of-entry, inline, and/or at point-of-use. Any installation and removal of filtration devices requires careful consideration by the WMP team. | Filters with a pore size of 0.2 μm or less that comply with industry standardized test methods can provide a barrier to transmission of *Legionella* [56]. |

**Table 1.** *Cont.*

| WQRSA Mitigation Strategy | Description | Notes |
|---|---|---|
| **Installing Physical Barriers** | A physical construction or partition barrier may be necessary to contain and prevent aerosolized water droplets from dispersing into the air, preventing exposure to waterborne pathogens via inhalation [48]. | Isolate building occupants and/or ventilation intakes from any device or equipment used to spray water, test or repair fixtures, operationalize misters, or equipment that contains a water reservoir. |
| **Recirculation and Hot Water Storage** | BWDS components including hot water recirculation systems and hot water storage present *Legionella* exposure risks and must be flushed if impacted due to a water-disruptive event including construction. | The volume of water flushed for these components should be considered in the BWDS flushing protocols. |
| **Equipment Installation, Cleaning, and Maintenance** | Building equipment and devices (e.g., ice machines, misters, showers, pressure washers) should be scheduled for installation or return-to-service in a timely manner (i.e., close to building occupancy) to avoid high water age, bacteria growth and spread [10,56,60,61]. | Operators must properly clean and maintain all building equipment using water per the manufacturer's recommendation prior to initial start-up and during routine operations. Similarly, remove and/or avoid premature installation of terminal fittings on fixtures (e.g., shower heads and hoses, aerators, faucet flow restrictors, screens, and filters in devices) before routine operations or in response to any water-disruptive event. |
| **Disinfection** | Disinfection is considered a highly effective method in the control of *Legionella* and can be a secondary, supplemental, or one-time (e.g., hyperchlorination) mitigation strategy [56]. | Important considerations when selecting disinfection method(s) include knowledge of the local municipal water or other water source, and the size and scope of the disruption event. |
| **Verification and Validation Testing** | Policies and procedures for analytical testing following water disruption events are the responsibility of the WMP team and must include the number of locations and the types of testing (e.g., physical, chemical, or microbial) to be performed [56,57]. | Each water disruption event is unique and requires an analysis of the size and scope of the event within the context of any existing or future WMPs to determine appropriate verification and validation test methods. |

## 3. Results

From the methods described, the authors developed a WQSRA tool using academic and school settings as the exemplar BWDS. Figure 3 depicts a summary of the WQSRA RDM outlining (1) the category (A, B, or C) and scope of BWDS disruption; (2) the building occupant risk groups (low, moderate, or high) to be impacted; and (3) the water risk mitigation level (RML-1, 2, or 3) to be implemented. Additionally, a brief description of the intent of each section of the WQSRA decision matrix is outlined below, as well as terms and definitions illustrated in Figures 4–7.

### 3.1. WQSRA Project Categories

3.1.1. Water Disruption Scope of Work

To assess the water disruption category (A, B, or C), the WMP team will identify the scope of work ranging from low disruptive incident or activity (Category A = inspection, maintenance, and noninvasive activities) to highly disruptive (Category C = major water-disruptive event or construction project). Further distinctions may include change of building function (e.g., renovation of office areas into higher-risk occupancy areas), shell area expansions, or acquisition and use of a tenant space/building for operations with unknown history of water quality and safety. The water disruption scope of work is to be evaluated in conjunction with water age to determine project category selection.

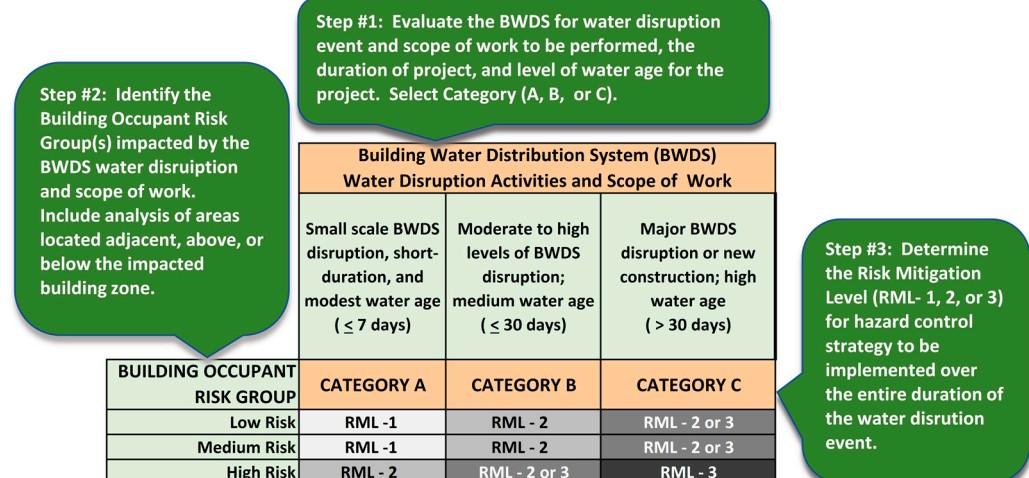

**Figure 3.** WQSRA risk decision matrix. Adapted with permission from Ref. [50]. 2022, Scanlon et al. (CC BY 4.0).

| | |
|---|---|
| **Instructions:** Evaluate the water disruptive event (Category A, B, or C). Categories are defined by the severity of the water disruptive event; the project scope of work to fix the issue; the extent of water age (i.e., stagnation), and the impact on building occupants. Contact the organization's Water Management Program Team or Safety and Risk Manager if any BWDS project activity needs clarification for completing the scope of work evaluation. | |
| **CATEGORY A** | **BWDS inspection, maintenance/repair, or small scale construction; creates minimal water disruption, and low to modest water age.**<br><br>Includes but not limited to:<br>Replacing fixtures and trim(s)<br>Replacing fixture "in-kind" (i.e., meaning 1:1 or like for like)<br>Replacing or installing fixtures in a localized area and may include work within wall cavities or ceiling areas.<br>Impact and risk is only to building users in the localized area of disruption (room, floor, or department)<br>Disaster event (e.g., inclement weather or leakage) allows containment within a small building area.<br>Water by fixture or area is shut down for $\leq$ 7 days (low to modest water age/stagnation) |
| **CATEGORY B** | **Work generates moderate to high BWDS disruption or removal of any fixed BWDS components or assemblies with medium water age.**<br><br>Includes but not limited to:<br>Plumbing work requiring multiple fixtures (existing, replacement or new)<br>Major water system components (boilers, heaters, water main, etc.)<br>Work in wall cavities or ceilings with major disruption to local and downstream occupied areas<br>Change of functional building space program (i.e. moving/changing room or dept. functions) in existing building<br>Disaster event is generally localized within the building yet impacts wide range of building occupants.<br>Water by fixture, component, or area is shut down $\leq$ 30 days (medium water age) |
| **CATEGORY C** | **Major BWDS disruption with extended periods of low and no building occupancy with high water age.**<br>Includes but not limited to:<br>Catastrophic events (public health disaster, major weather events, or emergency conditions)<br>Change in functional building space program (i.e., series of rooms, floors, or departments)<br>Tenant improvements (i.e., existing buildings, or tenant space within unoccupied buildings)<br>New shell and core buildings, additions, or expansions on campus (i.e. near existing academic environments)<br>Acquisition of building with unknown water quality / safety conditions<br>Infrastructure projects connecting to building water systems (i.e. underground piping, utility tunnels, etc.)<br>Water by fixture or area is not active (e.g., new start-up) or was shut down > 30 days (high water age) |

**Figure 4.** WQSRA water-disruptive event categories. Adapted with permission from Ref. [50]. 2022, Scanlon et al. (CC BY 4.0).

**Instructions:** Identify the Building Occupant Risk Group (i.e., students, faculty, staff, visitors, volunteers, or guests, etc.) and affected departmental areas. If more than one building occupant risk group will be affected, select the higher risk group / area of exposure. Contact the organization's Safety and Risk Manager if any risk group needs further clarification for relationships to the BWDS scope of work. The WMP team will need to know of any water reservoirs or fixtures in designated areas and risk from aerosolized water expsoure.

| Low | Medium | High |
|---|---|---|
| Administration areas | Cafeteria areas | Areas near cooling tower drift |
| Classrooms | Canteen/Nourishment | Athletic facilities (student, faculty, coaches) |
| Conference rooms | Food Prep Areas |   Whirlpools/spas |
| Laboratory (Dry) | Nutrition Stations |   Pools / special exercise equipment |
| Lobbies | Laboratory (Wet) |   Showers |
| Office areas | Laboratory (High-Priority Water) | Childhood / Pediatric venues |
| | |   childhood development centers |
| | |   daycare centers |
| | |   pediatric or family venues |
| | | Faculty/Guest overnight stay or lodging |
| | | Manufacturing or special curriculum |
| | |   Agricultural |
| | |   High Tech Computer Chip Processing |
| | |   Mining |
| | |   Telecscopic mirrors |
| | | Public campus areas with |
| | |   decorative water features |
| | |   shallow pools or bodies of water |
| | | Simulation/practicum education centers |
| | |   medical simulation centers |
| | |   nursing simulation centers |
| | | Seasonal venues |
| | |   Athletic stadiums |
| | |   Athletic field house / arena |
| | |   Entertainment / theater / auditoriums |
| | | Student dormitories |
| | | Student health and well-being |
| | |   student infirmaries |
| | |   outpatient clinics |
| | |   disability resources |

**Figure 5.** WQSRA building occupant risk groups. Adapted with permission from Ref. [50]. 2022, Scanlon et al. (CC BY 4.0).

3.1.2. Stratified Water Age Categories

Water age is known to degrade water quality and safety [9,56]. In non-healthcare settings, agencies frequently advise to flush and turn over BWDSs weekly [20]. Therefore, in analyzing risk for water-disruptive events in non-healthcare settings, we chose water age and stagnation cycles of up to 1 week (≤7 days), multiple weeks (≤30 days) [28], or after 1 month (>30 days) [28]. Water age is, therefore, stratified to define dormant water systems within each category description. Category A represents ≤7 days to define low water age projects to be performed with small-scale and short duration activities impacting the water system. Category B represents medium water age ≤30 days, which aligns with ANSI/ASHRAE Standard 188 Section 8.4.a.2.1, which requires repeat flushing if building occupancy is delayed 2 weeks but less than 4 weeks (after disinfection during start-up) [28].

Category C represents high water age >30 days, which represents significant time for low or no occupancy of existing buildings or start-up of new construction [28]. Extensive water age (>30 days) requires a project-specific commissioning plan.

When selecting the final WQSRA project category (A, B, or C), the WMP team must consider the water-disruptive activities, necessary scope of work, and length of system shutdown times contributing to water age. For example, a water heater replacement project that can be installed in fewer than 7 calendar days is not automatically determined to be a project for Category A selection. The WMP team would have to consider how many building occupants are impacted by the water heater replacement. If the water heater is a centralized building system, water disruption would likely impact building occupants throughout the building structure (i.e., not just a localized few rooms in the building). Assuming the water heater replacement project can be completed in <30 days, project Category B would be the appropriate selection. See Figure 4 for Water-Disruptive Event Category definitions.

| RISK MITIGATION LEVEL | **Instructions:** Review, finalize, and implement the selected WQSRA Risk Mitigation Level (RML) determined as appropriate for the BWDS water disruptive event, scope of work, and building occupant risk group. Contact the organization's Water Management Program Team and Safety or Risk Manager for clarification on individual hazard controls defined for the project duration.*1 |
|---|---|
| RML - 1 | 1) Establish enclosure to prevent aerosolized water (and potential pathogens) from dispersing into the environment.<br>　a) close door of area (i.e., room door, toilet/shower room door, etc.)<br>　b) install non-flammable visqueen or clear plastic sheeting or other approved vapor barrier for protection<br>　c) install isolation valve, backflow prevention device, or other piping isolation method<br>2) Prior to construction activities determine baseline measurements (i.e., temperature, residual disinfectant, pH, or other).<br>3) Flush fixture (hot) for a minimum 4 minutes; following flushing, collect water temperature using digital thermometer; perform the same minimum 4 minute flush (cold) and collect water temperature; record both measurements.<br>4) Collect residual disinfectant measurements (free or total) using a digital colorimeter instrument and record measurement.<br>5) Perform repair or replacement of plumbing components (i.e., plumbing fixture, trim, or other).<br>6) When water disruption activities are complete, and area is ready to return to service, flush the fixture for a minimum of 4 minutes hot, then 4 minutes cold. Take corresponding temperature and residual oxidant measurements. Repeat steps until measurements are the same or better than pre-existing conditions.<br>　a) Temperature: Hot water range [ 113$^o$F (45$^o$C) to 120$^o$F (48.9$^o$C)] and Cold water range [ $\leq$ 77$^o$F (25$^o$C)].<br>　b) Residual disinfectant range: Hot water = 0.20 ppm to 4.0 ppm and Cold water range = 0.20 ppm to 4.0 ppm<br>7) Report any odor, discolored water, flecks or floating debris at baseline or at work completion; none should be present.<br>8) Record information on organization's flushing form or in the project information system.<br>9) Facility / Construction staff to:<br>　a) leave barriers in place until all BWDS plumbing work is complete including flushing activities<br>　b) thoroughly clean and dry area(s) upon completion of project work<br>　c) remove barriers or seals in place<br>10) Environmental or construction cleaning services to perform routine cleaning before the area is occupied |
| *1 All mitigation measures (hazard controls) and associated numeric values (i.e. for temperature, residual disinfectant, pH, or other) need to be reviewed, coordinated, and implemented in context with the organization's on-going water management program. | |

**Figure 6.** WQSRA risk mitigation level 1 (RML-1). Adapted with permission from Ref. [50]. 2022, Scanlon et al. (CC BY 4.0).

### 3.2. WQSRA Building Occupant Risk Groups

After determining the category-specific BWDS disruption scope of work, the WMP team (e.g., facility manager, safety officer, industrial hygienist, among others) should identify impacted building occupants by building type, room function, and fixture type. Specifically for LD risk control, building occupant risk groups should be categorized from low to high to evaluate potential risk of exposure to aerosolized water. See Figure 5 for an example of building occupant risk group categories for academic and school settings.

| RISK MITIGATION LEVEL | **Instructions:** Review, finalize, and implement the selected WQSRA Risk Mitigation Level (RML) determined as appropriate for the BWDS water disruptive event, scope of work, and building occupant risk group. Contact the organization's Water Management Program Team and Safety or Risk Manager for health and safety for clarification on individual hazard controls defined for the project duration.*1 |
|---|---|
| RML - 2 | Review and/or perform ALL of RLM-1 risk mitigations and adjust for scale of project and:<br><br>11) Calculate water volumes for area of piping within BWDS project scope of work.<br><br>12) Perform flushing protocol [ _____ min. per day/ ___ days per week] on [circle day(s)] M, T, W, TH, F @ _______ fixtures in *occupied* areas adjacent to the project zone. Report on flushing form.<br><br>13) Perform flushing protocol [ _____ min. per day/ ___ days per week] on [circle day(s)] M, T, W, TH, F @ _______ fixtures in *unoccupied* areas within or adjacent to the project zone. Report on flushing form.<br><br>14) Obtain residual disinfectant and temperature readings post flushing activities 1 day per week in unoccupied and occupied areas at ___% of designated fixture locations as representative sample of fixtures to maintain adequate temperature and residual disinfectant levels. Report on fixture analysis form.<br><br>15) Review any disinfection (i.e. hyperchlorination) procedures to be performed with the Owner's Representative including location(s), method, schedule, and date establishing public BWDS potable water usage. Provide any reports of disinfection activities involving building water main (i.e. point-of-entry), or BWDS for hot and cold water lines.<br><br>16) If necessary, provide any temporary inline or point-of-use filtration for designated sinks, showers, or other fixtures or piping lines to reduce risk of exposure.<br><br>17) If necessary, provide any temporary auto-flushing devices at fixtures (i.e. sinks or toilets) at distal locations to pull water through system; set timing devices for [ _____ min. per hour / _____ times per day / _____ days per week]<br><br>18) Review installation for equipment with water reservoirs (i.e. ice machines or other) on the project and preventative maintenance prior to occupant start up including filter replacements.<br><br>19) Review options and finalize decision to perform analytical laboratory sampling for water quality contaminants (e.g., metals, general bacteria, or pathogens of interest). Use the risk of building occupants, baseline sampling, historical BWDS performance, regulatory requirements, or records from water management program as consideration. |
| RML- 3 | Consider using any RLM- 1 and 2 risk mitigation strategies and prepare a project specific WMP plan:<br><br>20) Contact the Building Owner's Representative for preparing a water commissioning project analysis<br><br>21) Conduct a project specific BWDS review of hazardous conditions associated with the water disruptive event. See Supplement S2: WQSRA Pre-Project Checklist.<br>  a) review site /civil construction or water disruption activity risk factors.<br>  b) review building design and construction or water disruption activity risk factors.<br><br>22) Based upon the Pre-project checklist prepare a project specific WMP plan for the BWDS per ANSI/ASHRAE 188 risk management process for WMPs.<br>  a) establish a WMP plan with scheduled milestones starting from the date of water disruption through water activation and toward a first-day of building operations.<br>  b) implement/operationalize project specific controls (i.e. protocols for flushing, temperature, and residual disinfentant readings).<br>  c) confirm WMP plan & operations with verification and validation.<br><br>23) Obtain Building Owner's Project Representative approval of the WMP plan, process, and documentation.<br><br>24) Implement the agreed upon WMP Plan for commissioning water quality and safety.<br><br>25) Obtain any authorities having jurisdiction (AHJs) and Building Owner's approval before initiating building occupant operations. |
| *1 | All mitigation measures (hazard controls) and associated numeric values (i.e., for temperature, residual disinfectant, PH, or other) need to be reviewed, coordinated, and implemented in context with the organization's on-going water management program. |

**Figure 7.** WQSRA risk mitigation levels 2 and 3. Adapted with permission from Ref. [50]. 2022, Scanlon et al. (CC BY 4.0).

### 3.3. Risk Mitigation Levels

Three water risk mitigation levels (RML-1, 2, or 3) contain a checklist of control methods and procedures to be undertaken throughout the phases of water disruption or construction until the project is complete and the BWDS returns to routine day-to-day operations. The RMLs build on one another. For example, a response to water disruptions or construction activity meeting the definition of RML-2 would include RML-1 controls such as baseline water parameter readings prior to the water disruption event, enabling comparison to post-event measurements to return the building water system to normal operating conditions. The RMLs were modified from the healthcare WMC-ICRA 4 × 4 cell

format [50] to a non-healthcare 3 × 3 cell format for academic institutions and schools. Figures 6 and 7 summarize the RML-1, 2, and 3.

Larger-scale water-disruptive events or construction projects with >30 days of dormancy or new start-up (i.e., Category C) should consider reviewing a pre-project checklist (PPC) (see Supplement File S2: WQSRA Pre-Project Checklist) for potential water disruption risk factors [19]. Following the PPC, a WMP team would create a project-specific water commissioning plan using the ANSI/ASHRAE 188 WMP method [28]. The water commissioning plan would be operationalized from the date of reactivation of the BWDS (i.e., water activation) in the BWDS and continue until the first day of routine BWDS operations. Controls (i.e., flushing protocols, temperature monitoring, and residual disinfectant monitoring) would be determined, implemented, and operationalized similar to those listed for RML-1, 2, and 3 but scaled for a larger BWDS and significant impacts from a water-disruptive event (e.g., flooding). The water commissioning plan and its implementation requires documented confirmation of the water management practice: (1) verification that the commissioning plan is being implemented as designed, and (2) validation that the commissioning plan, when implemented as designed, controls hazardous conditions throughout the BWDS. The WMP team would determine and be responsible for the water commissioning plan, supervising the implementation of all hazard controls, and project documentation. At the end of implementing the water commissioning plan, the WMP team will need to adjust and transition the plan toward an ongoing WMP and associated team members for continued building water system operations. A formal decision would be made by the building owner and any authority having jurisdiction (AHJ) concerning water quality and safety system approvals before initiating routine building operations and opening the building for public use.

*3.4. WQSRA Verification and Validation Testing*

Testing events may need to include water quality and safety baseline conditions (e.g., pre-event or some form of baseline testing) as well as testing for the same or improved conditions after the event or construction activities are completed (e.g., post-construction testing) [57]. Testing may involve water parameters such as temperature or residual disinfectant mentioned earlier [56]. Additionally, water chemistry (i.e., metals testing for elevated copper, lead, or other sediments) may be necessary related to older systems, systems with discolored water conditions, or systems with lead lines or components [20,62]. Furthermore, general bacteria testing (heterotrophic plate counts (HPC) or total heterotrophic aerobic bacteria (THAB) counts) may be used as an indicator of water quality [59]. Microbial testing should be considered based on the appropriate AHJ regulatory criteria, facility-specific microbial detection, epidemiology and disease surveillance history/records, and potential risk to building occupants [57].

Finally, as part of the commissioning process, the absence of Legionella should be verified prior to building occupancy, according to the NASEM 2020 report [12]. The CDC Toolkit for Controlling Legionella in Common Sources of Exposure [57] describes potable water test results of ≤1 colony forming unit per milliliter (CFU/mL) and cooling tower test results of ≤10 CFU/mL for Legionella as well controlled. However, readers should verify testing plans and protocols with their AHJ as the number and types of chemical or microbial testing or performance criteria may vary widely between AHJs [20,57,59].

## 4. Discussion

Higher education and school settings have always been subject to seasonal occupancy due to academic programs teaching using semester or quarter calendar systems. However, in our post-COVID-19 world, exposure risk from poor water quality will likely increase due to a wide variety of environmental and operational conditions. We discuss the need to have a WQSRA tool in the context of these environmental and operational changes that help building owners and water management professionals deal with BWDS disruptions and their importance to academic organizations.

### 4.1. Variable Building Occupancy (Seasonal, Low-Density, or Emergency Conditions)

The academic calendar results in weeks or months throughout the year with low or no building occupancy conditions (e.g., summer break, winter holiday, or spring break). During the pandemic response, students were sent home, while some institutions maintained limited working hours for faculty and staff. Liang et al. [5] tested 10 dormitory buildings at a university campus in Jiangsu, China, with 60 days of water stagnation during COVID-19 pandemic shutdowns compared to overnight stagnation and no stagnation conditions after reopening the buildings. The authors [5] reported statistically significant differences between 60-day water stagnation and overnight stagnation for both heterotrophic bacteria plate counts ($p = 3.1 \times 10^{-13}$) and *Legionella* spp. ($p = 0.0066$). Similarly, Ye et al. [3] tested 11 buildings located across three regional university campuses in Xiamen, China, with 40 days of water stagnation during the pandemic shutdowns. The authors [3] reported elevated zinc (>1000 µg/L) and iron (>300 µg/L), as well as poor residual disinfectant readings (<0.05 mg/L) and high turbidity (>1 NTU). The water samples were tested for microbes, and each building found detectable levels of *Legionella pneumophila*, *Salmonella* sp., *Shigella* sp., *E. coli*, *Pseudomonas aeruginosa*, and *E. faecalis*. The authors stated that once occupants returned and operations resumed routine building water use, excessive metals were lowered to typical limits in 3–7 days and within 1–2 months for microbes. Residual chlorine readings increased to >0.05 mg/L within 10–48 days. Turbidity returned to <1 NTU after 24 days of BWDS usage. This study examined long-term stagnation impacts on water quality and did not use flushing protocols. Instead, the research team tested water from 40 days of stagnation through 96 days post-occupancy with routine water building usage and reported findings. As institutions of higher education continue to utilize in-person, hybrid, and online instructional modalities [23], the likelihood of fluctuating occupancy in academic buildings and campuses will increase. Therefore, water age and stagnation become more problematic in these types of academic campus settings and likely demand more active water management controls, as well as confirmation of the BWDS's water quality and safety.

### 4.2. Increasing Water Management Policy Statutes

If WMPs become widely adopted as state laws, this would have a similar impact to codifying the 1974 Safe Drinking Water Act (SDWA) by the US EPA and would require dedicated resources to achieve and maintain compliance. Currently, most building owners have limited awareness of water quality and safety requirements, and the design and construction industries similarly are without training in this area of public health [12,19]. Building owners, facility management, safety and industrial hygiene teams, design professionals, construction workers, and commissioning agents need to have more awareness, tools, and training to become familiar with these standards and avoid creating unintended public health events. In 2015, ANSI/ASHRAE Standard 188 published the first uniform standard document for establishing building WMPs for US building owners for legionellosis risk management best practices [63]. The standard is updated every three years using a consensus approach among industry subject matter experts [28]. Under development is ANSI/ASHRAE Standard 514P: Risk Management for Building Water Systems: Physical, Chemical, and Microbial Hazards, which will be a similar standard forwater management addressing a wider range of water hazards to allow building owners to take a holistic approach to water management rather than focus solely on a singular pathogen such as *Legionella* [64]. These standards are voluntary until a legal jurisdiction makes compliance mandatory through legislation [28]. In 2015, within 60 days of publication, New York City adopted ANSI/ASHRAE Standard 188 Sub-Section 7.2 Cooling Towers and Evaporative Condensers as part of a public health response to a summer outbreak of legionellosis from cooling tower moisture "drift" that resulted in 120 disease cases and 12 deaths [65]. New York state was the first and remains the most robust state regulatory program for *Legionella* detection in the US. Other states are in the process of defining, passing, or rewriting water

quality and safety legislation, which, if passed, can have a large impact on a wide variety of key constituents and industries.

In September 2022, Governor Newsom vetoed the California Senate Bill (SB) 1144, which would have updated the California Safe Drinking Water Act with more robust requirements for water efficiency and quality for state buildings and public school buildings and required compliance no later than 1 January 2027 [66]. The bill's current language was determined to be too broad and cumbersome for implementation. A water efficiency and quality assessment report for each "covered building" was to have focused on (1) lead pipe identification and remediation; and (2) implementation of an ANSI/ASHRAE Standard 188 WMP for cooling towers [35]. A covered building was defined as any building owned and occupied or leased, maintained, and occupied by a state agency or public school building inclusive of charter school buildings. If revised and passed in the future, this would have significant implications for implementation at state university and school campuses, as well as for day-to-day BWDS management.

Similarly, if passed, Pennsylvania Senate Bill (SB) 1125 will effectively amend Title 27 Environmental Resources and Title 35 Health and Safety for LD prevention, including mandatory WMPs for all covered buildings and a restricted account for dedicated funding [37]. Covered buildings include any building as designated in ANSI/ASHRAE Standard 188 and all healthcare buildings under the US Federal reimbursement system from CMS. ANSI/ASHRAE Standard 188 [28] Section 5 suggests covered buildings would include any building meeting at least one of the following criteria: (1) open and closed-circuit cooling towers or evaporative condensers; (2) whirlpools and spas; (3) ornamental fountains, misters, or humidifiers; (4) multiple housing units with one or more centralized potable water heating systems; (5) buildings with 10 stories or higher (including below grade stories); (6) healthcare facilities with 24 h patient care services; (7) buildings housing or treating persons with special underlying disease status (e.g., burns, chemotherapy/cancer, solid organ transplant, or bone marrow transplant); (8) buildings treating immunocompromised patient types (e.g., renal disease, diabetes, chronic lung disease); and (9) buildings designated for housing occupants over age 65 years. Pennsylvania SB 1125 [37] also states that if construction is likely to create or elevate the presence of *Legionella pneumophila* in any covered building, the enforcing department would be required to direct the building owner to undertake appropriate infection control, prevention, and remediation measures to ensure public safety. In exchange for such compliance, the state of Pennsylvania suggested offering limited liability for civil damages and/or personal injury claims for actual or alleged exposure to *Legionella* or damages from exposure evolving into a LD case. This limited liability assumes compliance with the state's program, absent acts, errors, or omissions related to gross negligence, recklessness, willful misconduct, or intentional harm to others. At the time of publication of this article, SB 1125 is an open-status legislative initiative. The bill also requires a study to be performed analyzing the root cause and historical prevalence of *Legionella pneumophila* in public water systems that will be performed and presented to the State Assembly within one year of enacting this legislation.

In May 2022, Governor Pritzker signed for immediate implementation Illinois House Bill (HB) 4988, which modified Section 5 of the Environmental Protection Act [36,67]. Illinois HB 4988 [36] ensured that healthcare providers (e.g., hospitals, ambulatory surgical centers, and nursing homes) received notification from the community water system (CWS) of any (1) changes to disinfection method or routine practice impacting water quality; (2) planned events (e.g., construction) involving water distribution systems and building service mains; or (3) unplanned events (e.g., accidents, failed, or compromised water piping) impacting the healthcare facility's WMP and ongoing operations. The CWS is obligated to provide a 14-day prior notice of planned events, and a 2 h notice for unplanned events once a situation is identified, to the water regulatory authority. Although the Illinois legislation was directed solely at healthcare building owners, it illustrates a growing awareness among lawmakers of the dangers of water-disruptive events impacting water quality and safety.

Van Kenhove et al. [63] reviewed water management policies and *Legionella* regulations worldwide. Water management policies have been more common and implemented since the early 2000s in Europe, Asia, Australia, and the Middle East. The authors [63] reported a wide variety of guidance, standards, or code documents. The three guiding principles were: (1) management of BWDSs to monitor specific control locations; (2) avoiding high water age; and (3) maintaining water temperatures both hot (>60 °C) and cold (<25 °C) at distribution valves. The authors called for more uniform guidance documentation rather than relying on highly variable policy or enforcement criteria by country, province/state, or local authority having jurisdiction. Their aim of uniformity centered on climate change initiatives, and they articulated the need to agree on water and energy efficiency standards that do not produce unintended consequences of waterborne pathogen growth and spread in BWDSs.

*4.3. Green Building Design (GBD) Impact*

4.3.1. GBD and Educational Mission

GBD has become a cornerstone of academic campus development. In 2007, 685 colleges and universities signed a climate-neutral commitment pledge to reduce carbon emissions from campus buildings and increase research on climate, environment, and sustainability [24]. Historically, large academic research campuses are known to have high energy consumption buildings [24,68]. Academic campuses are considered small cities with a high impact on environmental quality due to a wide variety of building types and the high demand for goods and services [68]. Achieving sustainability initiatives through GBD is not just about reducing energy consumption; it has also been used for student recruitment, has become part of the academic curriculum, and impacts faculty hiring [69]. Green universities are ranked and rated for sustainable initiatives associated with infrastructure, energy, waste, water, and transportation that have environmental impact [68].

4.3.2. GBD and Water Quality

GBD has been associated with the spread and growth of waterborne pathogens in BWDSs. GBD was established to reduce energy and water usage and to minimize environmental impact from the building site [24,68]. A literature review by Allen and colleagues [70] examined articles tying GBD and human health outcomes to indoor environmental quality (IEQ). Of the 17 studies reviewed for either perceived or measured IEQ parameters, none mentioned indoor water quality or indoor air impacts from aerosolized water sources associated with the BWDS. Studies frequently discuss water conservation (i.e., efficiency and reduction in usage), without mention, measurement, or information on water quality or safety [12]. Green buildings are known to increase risk from *Legionella* and metals (e.g., lead) by increasing water age and lowering hot-water temperatures [12,26]. These water conservation and energy reduction initiatives subsequently diminish disinfectant residual and increase pipe corrosion, which can lead to metals leaching into the water supply [12]. Rhoads et al. [26] reported negligible residual disinfectant levels in GBD with decay up to 144 times faster in a green BWDSs with high water age.

As GBD has advanced over the last 20+ years, new water quality and safety threats are emerging related to off-grid or "net-zero" building designs [12]. A "net-zero" building design does not rely on a CWS for potable water or sewer piping for wastewater services. These designs use alternative water sources or reuse of water to reduce traditional potable water demand. Some techniques include water-saving devices, rainwater harvesting, water storage containment, and on-site gray water (i.e., reuse of water from clothes washers, sinks, or bathtubs) or black water (i.e., reuse of water effluent from toilets) [71,72]. These alternative methods of water supply carry obvious public health risks that need to be taken into consideration with water that is untreated or not treated to traditional regulatory standards. Even rainwater collection is not considered "clean" water since it can become contaminated by atmospheric conditions, rooftop materials, particulate matter, bird and animal feces, or other environmental debris [72]. Hamilton et al. [72] conducted a six-

month study testing water from roof-harvested rainwater storage tanks; each rainwater tank tested positive at least once during the study for a waterborne pathogen (e.g., *Legionella* spp., *Legionella pneumophila*, *Mycobacterium avium*, *Mycobacterium intracellulare*, *Pseudomonas aeruginosa*, or *Acanthamoeba* spp.).

### 4.3.3. GBD and Healthy Building Rating and Certification Programs

Criteria for GBD inclusive of a WMP would allow design professionals to comprehend the risk factors for waterborne pathogen growth and spread in BWDSs and balance public health protection with water conservation measures. The design and construction industries are not adequately trained to understand limitations of a GBD rating system and the specific science surrounding water quality and safety [12,50]. Often hazard control options such as flushing water systems are perceived by the building user/occupant as a waste of potable water [12]. To remedy this, separate from the US GBC LEED program, other environmental researchers [70,73] and building rating systems [74,75] have emphasized the fundamentals of a holistic approach to healthy buildings inclusive of water management practices. Allen and Macomber [55] created the nine foundations of healthy building measures inclusive of water quality. These authors recommended interactions between fields of expertise to manage IEQ and specifically suggested environmental health and industrial hygienists as necessary professionals in the evaluation of standardized healthy building certification programs and GBD. In terms of building rating systems, the Center for Active Design (CfAD) implemented FitWel®, an initiative started by the US CDC and General Services Administration [74]. The goal of FitWel® is to promote and certify health in buildings for all. In January 2020, FitWel® updated its certification V2.1 standard, policies, and protocols to include WMPs for building sites as well as listing water contaminant levels using both the US EPA and international WHO guidelines as the benchmark. Similarly, the International WELL Building Institute has published its updated WELL V2^TM standard for building health certification inclusive of water quality and safety [75]. The WELL V2^TM water category includes nine subsections addressing topics related to drinking water, water management, *Legionella* control, hygiene, and moisture control (e.g., mold). Both FitWel® and WELL V2^TM utilize US and international WHO references for WMPs and water safety plans to ensure alignment with the growing standards for building water management. Although this is a step in the right direction, the design and construction communities are not intimately familiar with best practices for WMPs [12], nor other traditional topics of environmental health or industrial hygiene [55]. A significant level of training will be necessary to course-correct away from decades of focusing solely on water efficiency project initiatives. The WQSRA tool can be a source of information and industry training in complement to these efforts. Before the GBD industry fully expands into the net-zero carbon emission design philosophy with more water conservation and reuse, it is critical that building design professionals, contractors, building owners, and building occupants be educated as to the minimum water quality and safety standards for building occupancy to establish appropriate project benchmarks and sustainable outcomes.

### 4.4. Education Campuses and Community Water Supply Systems

In 2022, there were 148,000 documented public water systems (PWSs) in the US, serving over 90% of Americans' tap water needs [76]. Water supplied by PWSs is regulated by the US EPA under the 1974 Safe Drinking Water Act (SDWA) and subsequent amendments. PWSs include community water systems (CWSs) that supply water to the same population year-round, non-transient non-community water systems (NTNCWSs), and transient non-community water systems (TNCWSs) that serve short-term users (e.g., campground facilities) [77]. A NTNCWS is defined as a system that regularly supplies water to 15 or more service connections and at least 25 of the same people at least six months out of the year. Hospitals, office buildings, and schools with their own water system are examples of NTNCWSs that comprise over 57% (85,000) of US PWSs.

Up to 10% of CWSs report health-based violations each year [77]. Federal standards are generally enforced at the state level; however, a lack of state inspection and monitoring to ensure compliance is a systemic limitation. The data are inherently flawed given that the accuracy of the information is dependent on PWSs' compliance with monitoring and reporting and the state primacy agency's efforts to review and report results to the US EPA Safe Drinking Water Information System (SDWIS), a publicly available database. Experts estimate a 38% level of violation underreporting [78].

Safe drinking water violations are more prevalent in CWSs serving smaller populations. Although minimal water quality and safety data are available specifically regarding NTNCWSs, historical data reflect that SDWA health violations are 14-fold more common in CWSs serving under 10,000 persons compared to larger systems [77]. Most NTNCWSs serve under 10,000 people. Violations were more frequent in government-owned utilities compared to privately owned and those with groundwater sources versus purchasing treated water from other utilities. Increased violations in rural, low-income, and minority populations may be due to the lack of financial resources and technical capacity to maintain system safety compliance needs and upgrades to meet ever-increasing regulatory standards [78].

State and local jurisdictions do not maintain a record of school water system providers. To find source water utility violations, a person would have to search individual utility records in the SDWIS. However, a study of 6974 public schools throughout California found that up to 24% were impacted by unsafe drinking water due primarily to bacterial and arsenic violations between 2003 and 2014 [79]. "Unsafe drinking water" was defined as a public water system that violated a primary maximum contaminant level (MCL) for contaminants regulated by the State Water Resources Control Board and reported in the state's Annual Compliance Reports. This study [79] predated lead and copper monitoring; thus, the percentage of schools with unsafe water would likely be much higher using contemporary standards. In rural, low-income regions, 1 in 3 schools was found to be impacted by unsafe water, further highlighting known disparity issues related to the supply of safe water, and 320 schools maintained their own NTNCWS. These systems were more likely to have water quality violations and recurring violations. In the California study, 52% (166) of schools served by NTNCWSs were impacted by unsafe drinking water [79]. The WQSRA tool can assist school authorities having to address changes to the BWDS due to elevated contaminant levels.

*4.5. Liability, Financial Risk, and Reputational Harm*

Academic institutions are in the unique position of providing goods and services, including potable water of sufficient quality and safety to students, faculty, staff, and visitors, and may therefore become liable to any of these groups for harm. When water-disruptive-events occur, concerns about campus water quality and safety can quickly develop into major public relations issues, requiring significant investment of time and resources to adequately address. Examples abound, but one timely case involved lead contamination in the potable water at the University of North Carolina (UNC). A 2010 study on the UNC campus revealed significant lead contamination exceeding the EPA Lead and Copper Rule Action Level of 15 ppb and highlighted the importance of water management, especially commissioning procedures, in new construction [80]. Similar to *Legionella*, water system design, water chemistry, low water demand, and high water age are important risk factors for exposure and require a "multifaceted approach that includes comprehensive testing and commissioning to remediate and mitigate problems" [80] (p. 75). Mitigation of the lead contamination described by Elfland et al. [80] cost over USD 30,000. Yet, these commissioning policies and practices seemingly did not completely resolve UNC's troubles related to risk of exposure to elevated lead contamination levels. In fall of 2022, UNC began offering blood lead testing to faculty, staff, and students through the University Employee Occupational Health Clinic and Campus Health at no cost to the individual, following revelations of high lead levels in water supplied at drinking fountains, sinks, and

ice machines after full campus operations resumed post-COVID [81]. The university's CWS, Orange Water and Sewer Authority, confirmed that the water leaving the drinking water treatment plant meets all regulatory requirements, including the lead standard, and that the issues at UNC are related to campus infrastructure [82]. While the costs of continued water and blood lead testing, plumbing upgrades, and staff time and resources have not been made publicly available, one can presume the costs of the 2022 campus-wide lead level remediation effort will far exceed the smaller-scale remediation in 2010. More challenging to quantify, of course, is the reputational harm associated with negative press coverage and social media postings that frequently accompany water-disruptive events [15,83]. Lack of access to water for drinking and hygiene can become a source of community outrage [84].

Similarly, recent events in Jackson, Mississippi, highlight the vulnerability of a CWS and the direct impact it can have on an academic institution. In July 2022, following decades of neglect, Jackson, Mississippi's CWS's service failures, poor water quality, and failure to maintain critical infrastructure forced the local community to live under an extended boil water advisory due to low water pressure throughout the service area [13,14]. Jackson State University, a historically black university in Jackson, was subsequently forced to switch to remote learning when drinking water, sanitation, and food services were not available on campus. Additionally, air conditioning was not available to cool indoor air amid the stifling late-summer Mississippi heat [14]. Primary and secondary school students were similarly affected by a return to remote learning and bottled water requirements, which presented financial and other burdens on this majority-black, economically depressed community [13]. Like the Flint, Michigan water crisis, the current Jackson, Mississippi challenges represent the culmination of years of maintenance delays, deferred investment, and staff shortages within the Jackson Water System [14]. This also included a change in water supply that may have affected water quality and increased lead contamination in potable water [13].

Another similar situation includes customers served by Trenton Water Works (TWW), who were notified by the New Jersey Department of Health (NJDOH) in October 2022 of the presence of *Legionella* in water samples collected in homes throughout the CWS service area [85]. Testing in thirty homes followed an investigation of an LD outbreak in the TWW service area in July 2022, which resulted in five cases, including one fatality, and "NJDEP's (New Jersey Department of Environmental Protection) finding of significant concerns with TWW's operations and management, including intermittent failures to fully maintain treatment processes, monitor water quality, employ adequately trained operating personnel, and invest in required maintenance and capital needs such as upgrades to aging infrastructure" [85] (p. 1). The NJDOH recommended homeowners act to decrease the risk of *Legionella* exposure in the home, including avoidance of high-risk activities (e.g., hot tubs), medical equipment maintenance, maintenance of hot-water temperature above 120 °F, cleaning of showerheads and aerators, and flushing, among others; they also provided similar guidance to building owners (e.g., healthcare, schools, commercial office) including cleaning and maintenance activities [85]. The ongoing compliance and operations issues at TWW led Governor Phil Murphy to direct the NJDEP to intervene and oversee operations, including operations and maintenance and capital investments. The TWW water crisis evinces the complex relationship between water purveyors, customers, and regulatory authorities in the control of *Legionella* in premise plumbing.

Although these incidents occur in community settings and may or may not directly involve an academic or school campus, a regional warning about chemical (e.g., lead) or microbial (e.g., *Legionella*) exposure can impact any institution's health, safety, or legal liability risk management programs. Flint, Michigan, UNC Chapel Hill, Jackson, Mississippi, and Trenton, New Jersey are but four examples of the US's fragile water infrastructure and supply. These examples demonstrate how a compromised water system adds to the challenge of water management. Ultimately, the health and safety of community residents (inclusive of academic and school campuses) rely on a consistent, affordable potable water supply beginning with the water purveyor and extending all the way through to the premise plumbing's final distribution points. Improved building water management, using

methods such as those presented in this paper, complements other recent research and policy activities aimed toward water justice [86].

*4.6. Limitations*

The WQSRA tool, tables, and definitions presented in the current study and by its authors are only an exemplar. The application of these tools requires any organization to review the contents for appropriateness and application in conjunction with the organization's WMP inclusive of policies for response to natural and man-made disaster events including construction. Every BWDS has unique characteristics which can create different outcomes. Any tools or policies used must be assessed by the responsible WMP team acting on behalf of the academic institution or school authority. Changes will be necessary for compliance with local, state, or federal policies regarding WMP legislation as well as alignment with the organization's existing WMP policy and implementation standards. All organizations assume the sole risk and full responsibility for implementation of such tools and practices, as well as the consequences of implementation in building environments. The authors make no representations or warranties about the suitability, completeness, reliability, legality, accuracy, or appropriateness of the information provided to reduce the likelihood of waterborne pathogens (e.g., *Legionella* or other) or the presence of water chemistry issues (e.g., elevated lead or copper) present in BWDSs. The authors recommend appropriate training for all WMP teams and their representatives to reduce the likelihood of emerging disease cases, injuries, or deaths that may occur from BWDSs and improper implementation of WMPs.

**5. Conclusions**

Protection of students, faculty, staff, and visitors from a potentially unsafe BWDS is essential during disaster events and construction activities [1,5,14]. A BWDS risk management plan is recommended inclusive of water-disruptive events [5,14]; however, prior tools have not focused on the approximately 5900 US colleges and universities and the 128,000 US elementary and secondary schools who may need to perform a building water quality and safety risk assessment during a disaster [42]. Our recommendation is for each academic organization to review and develop a policy leveraging the methods demonstrated within the WQSRA tool and customize their efforts to the site-specific/organizational WMP to address five potential gaps:

- Reduce the likelihood of a disease case, injury, or death from a water-disruptive event due to water quality or safety issue emerging from the BWDS;
- Reduce the likelihood of a water quality or safety issue from a water-disruptive event in a BWDS undergoing any project maintenance, repairs, or involving construction activities;
- Reduce the likelihood of unintended health consequences in BWDSs from design professionals utilizing building or certification systems focusing primarily on green building design initiatives without consideration of water quality and safety;
- Improve regulatory alignment for building owners with emerging new state policies requiring risk mitigation for water chemistry or waterborne pathogen growth and spread in BWDSs;
- Extend public health training to include comprehensive WMP methods for catastrophic events, water disruption, and construction activities.

**Supplementary Materials:** The following supporting information can be downloaded at: https://www.mdpi.com/article/10.3390/buildings13040921/s1, Supplemental File S1: WQSRA Building Water Distribution System for Academic Institutions or School Settings (non-healthcare settings); Supplemental File S2: WQSRA Pre-Project Checklist.

**Author Contributions:** Conceptualization, S.C.G. and M.M.S.; methodology, S.C.G., M.M.S. and K.A.R.; formal analysis, S.C.G., M.M.S. and K.A.R.; writing—original draft preparation, S.C.G., M.M.S. and K.A.R.; writing—review and editing, S.C.G., M.M.S. and K.A.R.; visualization, M.M.S.; project

administration, M.M.S. and S.C.G. All authors have read and agreed to the published version of the manuscript.

**Funding:** This research received no external funding.

**Data Availability Statement:** All data are contained within the article or the Supplemental Materials.

**Conflicts of Interest:** The authors declare no conflict of interest.

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
