# Peer review of "Managing Building Water Disruptions in a Post-COVID World: Water Quality and Safety Risk Assessment Tool for Academic Institutions and School Settings"

_buildings, doi:10.3390/buildings13040921_

Round 1

Reviewer 1 Report

This study presents the Managing Building Water Disruptions in a Post-COVID  World: Water Quality and Safety Risk Assessment Tool for Academic Institutions and School Settings. The study is well written, however, a few comments need to be addressed before accepting for publication.

1. Introduction needs to be rewritten again to be comprehensive.

2. The novelty of the paper needs to be highlighted better.

3. I recommend removing the subsections 1.1, 1.2, 1.3.... and merge all of them under the introduction. Summrize only the paragraphs related to the paper topic.

4. Figure 2 should be in Table forms.

5. Figure 3 should be moved to literature review.

Author Response

Response from Authors to Reviewer 1

Reviewer:  This study presents the Managing Building Water Disruptions in a Post-COVID  World: Water Quality and Safety Risk Assessment Tool for Academic Institutions and School Settings. The study is well written, however, a few comments need to be addressed before accepting for publication.

  1. Introduction needs to be rewritten again to be comprehensive.

Authors: Thank you for your comment.  Based on the collective peer-review comments, we have modified the Introduction to better explain the reason for various background topics to be included in the current study, as part of our research into creating a building water management tool for educational settings.  This was accomplished in context to your other comments.  See our response under Items 2 and 3 below.

  1. The novelty of the paper needs to be highlighted better.

Authors: Thank you for your comment.  We have inserted a new paragraph to describe the novelty of our manuscript in support of our aims of the research which were already stated.  Other text was added surrounding the novelty portion as suggested by other reviewers.  Please see lines 253 to 279 for context.  Here is a summary of the revised and inserted text (without citations) at the end of the introduction:

To summarize, although there has been significant worldwide water quality research for utility infrastructure, environmental water pollution, water hydraulic behavior, and municipal water delivery systems for drinking water, there are far fewer examples of water research, tools, and standards focused on the building owner and their respective water management team members who are often the sole decision makers responsible to manage water quality for building occupant usage. In recommending WMPs for all public buildings, the NASEM report suggested creating derivative tools or products focusing on non-healthcare buildings to further advance water management best practices to a wider building owner audience. The novelty of this study is represented by the building type, academic institutions and school settings, and the simplicity of the tool for implementation during a disastrous event impacting water quality and safety. First, educational settings are a different building occupancy type (i.e., non-healthcare) and need risk assessment tools addressing environments used by children, minors, and young adults, as well as faculty, staff, visitors, and other community members. Second, the building owner’s facility management personnel often get involved in a water disruptive event, given their knowledge about the BWDS physical components (i.e., fixtures, piping, and water heaters), yet they may not fully understand or be trained in building water management or basic water science. The building owner’s representatives need tools to rapidly evaluate the water disruptive event, establish team consensus, and subsequently communicate instructions to other stake holders about necessary modifications to the BWDS to re-establish water quality for on-going operations. The purpose of the current study is to fill this gap in current water management practices by developing a water quality and safety risk assessment (WQSRA) tool that will: 1) assist higher education and school-based water management teams with evaluating water quality challenges at academic campuses and school environments during water disruptive events, and 2) increase awareness and reduce potential risk to all building occupants (e.g., students, faculty, staff, and visitors).

  1. I recommend removing the subsections 1.1, 1.2, 1.3…. and merge all of them under the introduction. Summarize only the paragraphs related to the paper topic.

Authors:  Thank you for your comment.  In reviewing the structure of the Introduction, we realized we lacked clarity to transition between the topics stated in 1.1, 1.2, 1.3, etc. in the manuscript.  Rather than remove the headings, merging the text, and eliminating key background information, we prefer to maintain the subtitles which act as clear subject identifiers in the background to enhance readability and comprehension in a longer Introduction section. This is important since: a) this manuscript is being published in a journal focused on building systems (i.e., typically addressing building system performance criteria) with an audience potentially unfamiliar with water quality research, and b) MDPI journal structure does not have a separate background section into which we might place all this information; therefore, we meet this need by maintaining the edited and improved Section 1 Introduction with sub-topic identifiers.  Specifically, to improve the manuscript and address your comment, we modified the Introduction and deleted any verbose language where appropriate.  We added the following text on lines 64- 74.

Similar to the impacts observed during the COVID-19 pandemic, academic institutions and schools are located in community settings which are increasingly vulnerable to surrounding environmental conditions and disaster events. Water quality and disaster events are often interchangeably linked. As we have seen with the Flint, Michigan and Jackson, Mississippi water crises, as well as the East Palestine, Ohio train derailment, any community and its local educational system can be critically impacted based on compromised water resources or infrastructure. As background analysis for the present study to create a tool for water quality risk assessment, we reviewed five key areas related to educational settings including: 1) water disruptive events; 2) green building design and water system challenges; 3) municipal water supply changes; 4) water management policies for BWDS; and 5) risk decision matrix concepts.

We hope this explanation and updated approach meets with your satisfaction.

  1. Figure 2 should be in Table forms.

Authors: Thank you for your comment.  Figure 2 is a collection of diagrams illustrating risk decision matrix chart formats.  These are not tables in the typical sense with data.  MDPI typically wants “tables” in an interactive software editing format which would not allow us to keep the three chart images collectively represented as graphic diagrams (i.e., Figures).  However, to address your comment we edited the text (lines 200 to 206) to refer to these images without using words such as table or tabular to avoid any confusion to the reader.  If there are any further format issues, we will gladly resolve that with the academic/desk editor staff during final formatting prior to publication. 

We hope this explanation meets with your satisfaction.

  1. Figure 3 should be moved to literature review.

Authors: Thank you for your comment.  Figure 3 is a summary of our results.  This is an original diagram summarizing how a new risk decision matrix (3 x 3 format) would be depicted based on our findings.  From our point of view, it would be contrary to research practice to insert our summary of results into the Introduction and background section of the paper.  Figure 2, as stated above, contains the charts/illustrations of the background on risk decision matrix examples. 

Reviewer 2 Report

Dear authors,
thanks for your contribution firstly.
The paper provides a Water Quality and Safety Risk Assessment (WQSRA) tool to address gaps in building water management for academic institutions and school settings.

The abstract briefly summarizes the purpose of the paper and overall the article is well structured.
At the end of the introductory paragraph, I suggest that you better highlight the scientific novelty of your study. Even if the aim of the study is well explained it is important to highlight what the additional / new contribution to the research is.
 It would be useful, as well as interesting, to insert at least one introductory paragraph that deals with the efforts made by research to improve water quality, not only strictly related to your study type but in general. To this end, I suggest the inclusion of the following studies that could give to value the problem treated in your study (doi: 10.3390 / s20123432, https: // doi.org/10.3390/w13070934 , https://doi.org/10.3390/w14172707); in fact, they deal with the water quality, it is useful to give an overview of the importance of theme that  interests entire water system "from the cradle to the grave".

In conclusion, I believe that the paper includes solid content, but some aspects need to be improved, improving them this manuscript can have its own value and impact.

I hope that these recommendations are helpful to the authors and wish good luck for the further reviewing process.

Author Response

Response from Authors to Reviewer 2

Reviewer:  The paper provides a Water Quality and Safety Risk Assessment (WQSRA) tool to address gaps in building water management for academic institutions and school settings.

Authors:  Thank you for providing us comments on our work. We as authors added numbers to the additional comments just to organize our responses to you.

  1. The abstract briefly summarizes the purpose of the paper and overall the article is well structured.

Authors:  Thank you for your comment and support of our work.  Based on another reviewer’s comments we have provided slight modifications to the abstract. 

  1. At the end of the introductory paragraph, I suggest that you better highlight the scientific novelty of your study. Even if the aim of the study is well explained it is important to highlight what the additional / new contribution to the research is.

Authors:  Thank you for your comment.  We have inserted a paragraph to describe scientific novelty in support of our aims for the research.  The text was provided as part of addressing your comment in Item #3 below.  See Item #3 for a combined response. 

  1. It would be useful, as well as interesting, to insert at least one introductory paragraph that deals with the efforts made by research to improve water quality, not only strictly related to your study type but in general. To this end, I suggest the inclusion of the following studies that could give to value the problem treated in your study (doi: 10.3390 / s20123432, https: // doi.org/10.3390/w13070934 , https://doi.org/10.3390/w14172707); in fact, they deal with the water quality, it is useful to give an overview of the importance of theme that  interests entire water system "from the cradle to the grave".

Authors:  Thank you for your comment.  We read the two articles suggested and included them in our reference/citations.  In another reviewer’s comment, we were asked to trim the Introduction in contrast to this suggestion to expand it by a paragraph. As a compromise between the two suggestions and to keep the manuscript focused on building water research, we have added a summary paragraph (see lines 253 – 279) about water quality research in the context of our work.  The text (without citations) is summarized here:

To summarize, although there has been significant worldwide water quality research for utility infrastructure, environmental water pollution, water hydraulic behavior, and municipal water delivery systems for drinking water, there are far fewer examples of water research, tools, and standards focused on the building owner and their respective water management team members who are often the sole decision makers responsible to manage water quality for building occupant usage. In recommending WMPs for all public buildings, the NASEM report suggested creating derivative tools or products focusing on non-healthcare buildings to further advance water management best practices to a wider building owner audience. The novelty of this study is represented by the building type, academic institutions and school settings, and the simplicity of the tool for implementation during a disastrous event impacting water quality and safety. First, educational settings are a different building occupancy type (i.e., non-healthcare) and need risk assessment tools addressing environments used by children, minors, and young adults, as well as faculty, staff, visitors, and other community members. Second, the building owner’s facility management personnel often get involved in a water disruptive event, given their knowledge about the BWDS physical components (i.e., fixtures, piping, and water heaters), yet they may not fully understand or be trained in building water management or basic water science. The building owner’s representatives need tools to rapidly evaluate the water disruptive event, establish team consensus, and subsequently communicate instructions to other stake holders about necessary modifications to the BWDS to re-establish water quality for on-going operations. The purpose of the current study is to fill this gap in current water management practices by developing a water quality and safety risk assessment (WQSRA) tool that will: 1) assist higher education and school-based water management teams with evaluating water quality challenges at academic campuses and school environments during water disruptive events, and 2) increase awareness and reduce potential risk to all building occupants (e.g., students, faculty, staff, and visitors).

We hope this explanation and updated approach meets with your satisfaction.

  1. In conclusion, I believe that the paper includes solid content, but some aspects need to be improved, improving them this manuscript can have its own value and impact. I hope that these recommendations are helpful to the authors and wish good luck for the further reviewing process.

Authors:  Thank you for your comment and support of this manuscript.

Round 2

Reviewer 2 Report

Dear Authors,

the requested changes have been made. Therefore, in my opinion, the paper is ready for the to publication.